# Comparison of Three Different Diagnostic Assays for Fibroblast Growth Factor-23 (FGF-23) Measurements in Cats: A Pilot Study

**DOI:** 10.3390/ani13111853

**Published:** 2023-06-02

**Authors:** Sandra Lapsina, Nicole Nagler, Simon F. Müller, Annette Holtdirk, Tanja Kottmann, Elisabeth Müller, Jennifer von Luckner, Ingo Schäfer

**Affiliations:** 1LABOKLIN GmbH and Co. KG, Steubenstraße 4, 97688 Bad Kissingen, Germany; lapsina@laboklin.com (S.L.); nagler@laboklin.com (N.N.); sf.mueller@laboklin.com (S.F.M.); mueller@laboklin.com (E.M.); luckner@laboklin.com (J.v.L.); 2Dr. med. Kottmann—Clinical Research Organization, Beverstraße 64, 59077 Hamm, Germany; a.holtdirk@cro-kottmann.de (A.H.); tk@cro-kottmann.de (T.K.); 3AniCura Ahlen, Bunsenstraße 20, 59229 Ahlen, Germany

**Keywords:** feline, chronic kidney disease, phosphate, clinical pathology

## Abstract

**Simple Summary:**

Fibroblast growth factor-23 (FGF-23) is used to monitor chronic kidney disease in humans. The value of FGF-23 in cats is largely unknown. The aim of this study was to compare three different FGF-23 diagnostic assays and to assess their correlation with renal function parameters in cats. Four groups were formed retrospectively on the basis of creatinine concentration classification according to the IRIS guidelines. Measurements were performed in 40 cats in total. The Kainos ELISA assay (FGF-23 ELISA Kit, Kainos Laboratories, Tokyo, Japan) showed the highest correlations for all study groups. Individual assay-based reference values should be established to make a reliable interpretation of FGF-23 levels possible to diagnose or monitor feline CKD.

**Abstract:**

Fibroblast growth factor-23 (FGF-23) is a phosphaturic hormone used to monitor chronic kidney disease (CKD) in humans. The aim of this pilot study was to compare three diagnostic assays and to assess how the results correlate with parameters of renal dysfunction in cats. Four groups of 10 cats each were formed retrospectively according to creatinine, based on IRIS staging. FGF-23 was measured using two different ELISAs (MyBioSource and Kainos ELISA FGF-23 Kit) and an automated assay on the DiaSorin Liaison platform. Measurements were performed in 40 cats. Spearman’s rank correlation coefficient showed a strong correlation between the Kainos and DiaSorin assays (ρ = 0.742/*p* < 0.001) and a low correlation (ρ = 0.443/*p* = 0.005) between the Kainos and MyBioSource assays. The measurements with the Kainos assay strongly correlated with urea (ρ = 0.835/*p* < 0.001) and creatinine (ρ = 0.764/*p* < 0.001), and moderately correlated with SDMA (ρ = 0.580/*p* < 0.001) and phosphorus (ρ = 0.532/*p* < 0.001). The results of the MyBioSource and DiaSorin assays only showed a moderate correlation with urea (ρ = 0.624/0.572) and creatinine (ρ = 0.622/0.510) concentrations (*p* = 0.001 each). The Kainos assay showed the strongest correlation (ρ = 0.806) with the various creatinine concentrations according to the IRIS, followed by the MyBioSource (ρ = 0.663/*p* < 0.001) and DiaSorin assays (ρ = 0.580/*p* < 0.001). Overall, the Kainos assay demonstrated the best correlations with both biomarkers and various creatinine concentrations according to the IRIS. Individual assay-based reference values should be established to make a reliable interpretation of FGF-23 levels possible to diagnose or monitor feline CKD.

## 1. Introduction

Chronic kidney disease (CKD) is one of the most common biochemical abnormalities in cats. Its prevalence is 2–4% overall [1,2] but can be as high as 30–40% in cats over 10 years of age [3,4]. CKD in cats is often idiopathic, but possible risk factors include frequent vaccination [5], dental disease [5,6], and a history of cystitis or general anesthesia within 12 months [6]. CKD is a progressive disease with a highly variable timeframe for progression to end-stage disease. The prognosis is based on staging according to the International Renal Interest Society (IRIS) guidelines, which include serum creatinine concentration, blood pressure measurements, and urine protein/creatinine ratio [7]. The modified IRIS guidelines also include serum symmetric dimethylarginine (SDMA) concentration [8].

Newer renal biomarkers in cats, such as cystatin C, SDMA, and Fibroblast growth factor (FGF)-23, measure the glomerular filtration rate (GFR) and/or deranged metabolism.

Cystatin C is not a reliable biomarker of CKD in cats [9], as reports on its changes with CKD staging or even presence have been greatly inconsistent [10,11,12]. It has long been thought that there are higher levels in cats with CKD than in healthy controls; however, studies have shown no difference in cystatin C concentrations among various IRIS stages of CKD [11] nor between affected cats and healthy controls [13,14].

SDMA is known to be a sensitive biomarker. It allows a decreased glomerular filtration rate (GFR) to be detected in an earlier stage than creatinine (mean 40% reduction in the GFR in contrast to 75%) [15] and is not affected by muscle mass [16,17,18]. Nevertheless, breed-related differences in baseline SDMA concentration have been documented [18], and there are limited data regarding its specificity [19,20]. Persistently increased SDMA concentration may indicate early CKD [8].

FGF-23 is a phosphaturic hormone produced by osteocytes and osteoblasts. It is responsible for phosphorus and calcitriol homeostasis [21,22]. FGF-23 regulates serum phosphate levels by increasing renal phosphate excretion and decreasing the synthesis and degradation of calcitriol. An increase in FGF-23 concentration promotes urinary excretion of phosphorus and decreases its absorption in the intestines [23,24,25]. In addition, FGF-23 acts on the parathyroid gland and reduces both the synthesis and secretion of parathyroid hormone [26]. In humans with CKD, serum FGF-23 concentrations rise exponentially as renal function declines long before a significant increase in serum phosphate or parathyroid hormone concentration can be detected [26,27,28]. Additionally, FGF-23 concentration is negatively correlated with the estimated GFR in humans [28]. Hyperphosphatemia and increased calcitriol concentration in the blood stimulate FGF-23 production.

Several studies indicate that FGF-23 may be a promising early biomarker of phosphate derangement in feline CKD [29,30,31,32,33]. FGF-23 might have diagnostic relevance in the early detection of both CKD and phosphate derangement in cats with CKD. Disruption of phosphate homeostasis may develop in the early stages of CKD, even before azotemia or hyperphosphatemia occurs. FGF-23 concentration is significantly higher in cats with azotemic CKD than in healthy animals and significantly increases with the progression of CKD [29,30]. However, the true diagnostic potential of FGF-23 in cats still needs to be investigated.

Several studies investigating FGF-23 concentration in cats have been performed so far. Different diagnostic quantitative enzyme linked immunosorbent assay (ELISA) kits have been used for this purpose: Four of these studies used the FGF-23 ELISA kit (Kainos Laboratories, Tokyo, Japan) [29,31,33,34]; two further studies used MyBioSource ELISA (MyBioSource, San Diego, CA, USA) [30,35]; and in another one, MedFrontier FGF23 ELISA (Minaris Medical Co., Tokyo, Japan) [36] was used. As knowledge regarding the performance of different diagnostic assays for FGF-23 concentration measurements in cats remains limited, the aim of this study was to compare three diagnostic assays for the measurement of FGF-23 concentration and to evaluate the correlation of this parameter with other parameters involved in renal dysfunction in cats.

## 2. Materials and Methods

This retrospective study was conducted in the commercial laboratory LABOKLIN (Bad Kissingen, Germany). Surplus blood serum from routine samples was used for all measurements. The original samples were submitted to the LABOKLIN commercial laboratory (Bad Kissingen, Germany) by veterinarians in Germany in December 2022. According to the terms and conditions of the LABOKLIN laboratory, as well as the RUF-55.2.2.2532-1-86-5 decision of the government of Lower Franconia, no special permission from animal owners or the animal welfare commission is needed for additional testing on residual sample material once diagnostics are completed. The general guidelines of the laboratory recommend taking blood samples in cats after animals have fasted overnight. Biochemical parameters were evaluated immediately after the overnight shipment of serum samples to the laboratory. For FGF-23 analysis, samples were stored frozen for a maximum of 5 working days at −20 °C.

Four study groups were selected based on serum creatinine concentrations reported for IRIS CKD stages 1–4 [8]. Cats were retrospectively assigned to these groups based on their serum creatinine concentrations, but groups were not fully equivalent to IRIS stages 1–4.

FGF-23 concentration was measured using two different quantitative ELISA assay kits (MyBioSource, San Diego, CA, USA; FGF-23 ELISA Kit, Kainos Laboratories, Tokyo, Japan) and the LIAISON FGF 23 kit for intact FGF-23 on the Liaison platform developed by DiaSorin (Saluggia, Italy) according to the manufacturers’ guidelines. The manufacturers’ specifications provide a minimum detection limit of 3 pg/mL and a quantification range of 3–800 pg/mL for the Kainos ELISA assay [37], a detection range from 15.6 to 1000 pg/mL for the MyBioSource ELISA assay [38], and a 5 pg/mL detection limit and a 6.5 pg/mL quantification limit for the DiaSorin assay.

Renal function parameters such as creatinine, SDMA, urea, phosphorus, calcium, potassium, and sodium were measured using Cobas 8000 (Roche, Mannheim, Germany).

Descriptive statistical analysis was performed using SPSS for Windows (version 28.0; International Business Machines Corporation, Armonk, NY, USA). *p* < 0.05 was considered statistically significant. The Shapiro–Wilk test was used for the assessment of the normal distribution. The Kruskal–Wallis test was used to calculate statistical significance among the study groups. Bonferroni correction was applied where indicated. Spearman’s rank correlation coefficient (ρ) was also calculated. ρ < 0.2 was classified as a very low correlation; ρ = 0.2–0.5, as a low correlation; ρ > 0.5–0.7, as a moderate correlation; ρ > 0.7–0.9, as a strong correlation; and ρ > 0.9, as a very strong correlation. Bland–Altman plot analysis was used for comparison of FGF-23 concentrations measured using the three diagnostic assays. The limits of agreement (LAOs) were calculated using the following equation: LOA = the mean difference ± 1.96 × standard deviation of the difference.

To assess both intra-assay and inter-assay precision, the coefficient of variation (CV) was calculated. For intra-assay precision, five FGF-23 concentration measurements in samples at three different concentrations (low, moderate, and high) within a single run were taken. For inter-assay precision, FGF-23 concentration was measured at three different concentrations (low, moderate, and high) in the same sample once a day for five consecutive working days. FGF-23 concentrations were graded according to the results provided by the Kainos ELISA assay as follows: low, 70–120 pg/mL; moderate, 125–580 pg/mL; and high, 3200–3600 pg/mL.

## 3. Results

Forty cats were included in the study. Serum samples for FGF-23 concentration measurements with all three diagnostic assays were available for 38 cats (Kainos assay, *n* = 40; MyBioSource and DiaSorin assays, *n* = 38 each) (Figure 1). The sample volume available from the remaining two cats was insufficient for FGF-23 measurements with the MyBioSource and DiaSorin assays. The results of SDMA measurements were available for 31 cats, and the results of the other biochemical parameters were available for all 40 cats.

The following additional parameters were collected: age (available for 29/38 cats or 76.3%; mean of 10.6 years, median of 12.0 years, standard deviation of 5.8 years, minimum of 1 year, and maximum of 18 years; see Table 1), breed (available for 35/38 cats or 92.1%; the majority were European Shorthair cats (*n* = 25, 71.4%), followed by British Short-/Longhair (*n* = 4, 11.4%), Maine Coon (*n* = 2, 5.7%), mixed breed (*n* = 2, 5.7%), Carthusian (*n* = 1, 2.9%), and Persian (*n* = 1, 2.9%) cats), and sex (available for 35/38 cats or 92.1%; a total of 14/35 cats or 40.0% were male, where intact *n* = 4 and castrated *n* = 10; a total of 21/35 cats or 60.0% were female, where non-spayed *n* = 9 and spayed *n* = 12).

The Shapiro–Wilk test showed a non-normal distribution for FGF-23 concentration in all three diagnostic assays (*p* < 0.001 each). This was the same after log-transforming the data prior to statistical analysis (*p* < 0.018 each). Therefore, Bland–Altman plot analysis was not a suitable test for comparing the performance of the different assays, as the FGF-23 data of the means among the assays were not normally distributed.

The renal function parameters also significantly deviated from a normal distribution (*p* < 0.001 each), except potassium (*p* = 0.680) and calcium (*p* = 0.275). A statistically significant difference among the study groups was noted for SDMA, phosphorus, and urea concentrations (*p* < 0.001 each) but not for calcium (*p* = 0.862), sodium (*p* = 0.709), or potassium (*p* = 0.231) (Table 1).

FGF-23 concentration differed significantly among the study groups according to all three assays (Table 1). None of the three diagnostic tests included in the study (Kainos ELISA, MyBioSource, and DiaSorin) showed a statistically significant difference in FGF-23 levels between study groups I and II. However, all three assays showed significantly different FGF-23 concentrations between groups I and IV. With the Kainos ELISA assay, a statistically significant difference in FGF-23 concentration measurements was found between study groups I and III, II and IV, and I and IV (Table 2). With the MyBioSource assay, a statistically significant difference was seen between study groups I and III, and I and IV. With the DiaSorin assay, there was a statistically significant difference between study groups I and IV, and II and IV (Table 2).

The Kainos ELISA assay showed a strong (ρ = 0.730, *p* < 0.001) correlation with the DiaSorin assay and a moderate (ρ = 0.560, *p* < 0.001) correlation with the MyBioSource assay. The MyBioSource ELISA assay demonstrated a moderate correlation with the Kainos ELISA assay (ρ = 0.560, *p* < 0.001) and a low correlation with the DiaSorin assay (ρ = 0.423, *p* = 0.010). The DiaSorin assay showed a strong correlation with the Kainos ELISA assay (ρ = 0.730, *p* < 0.001) and a low correlation with the MyBioSource assay (ρ = 0.423, *p* < 0.001) (Table 3 and Figure 1).

Only the Kainos ELISA showed statistically significant differences in FGF-23 levels between study groups I and III, I and IV, and II and IV (Table 2, Figure 2). There was a strong correlation between FGF-23 concentration measured with the Kainos ELISA diagnostic assay and urea (ρ = 0.835) or creatinine (ρ = 0.764) measurements (*p* < 0.001 each). The results of this assay also showed a moderate correlation with SDMA (ρ = 0.580) and phosphorus (ρ = 0.532) (*p* < 0.001 each), as well as a low correlation with sodium concentration (ρ = 0.332, *p* = 0.036) (Table 3). The FGF-23 concentration measured with the DiaSorin assay had a very strong correlation with creatinine levels (ρ = 0.950, *p* < 0.001), a moderate correlation with urea (ρ = 0.572, *p* < 0.001), and a low correlation with phosphorus (ρ = 0.420, *p* < 0.001) and sodium (ρ = 0.337, *p* = 0.039). The results of the MyBioSource ELISA assay only showed a moderate correlation with urea (ρ = 0.624) and creatinine (ρ = 0.510) concentrations (*p* < 0.001 each).

The results given by the Kainos ELISA were the most promising in terms of differentiating among the various groups in the study (Table 2) and correlating with other renal function tests, such as creatinine, SDMA, phosphorus, urea, and sodium (Table 3). Therefore, precision measurements were performed for this assay. The intra-assay precision showed CVs of 12.54%, 4.17%, and 4.19%, while inter-assay precision values were 9.2%, 9.96%, and 8.17% for samples with low, moderate, and high FGF-23 levels, respectively.

## 4. Discussion

There is very limited knowledge regarding FGF-23 concentration measurements obtained using individual diagnostic assays in cats or how these assays may compare. The majority of studies to date have tested the Kainos ELISA [29,31,32,33,34] and MyBioSource ELISA assays [30,35], both of which are quantitative ELISA tests that must be operated manually. This study additionally tested the DiaSorin assay, which is an automated assay running on the Liaison platform and thus might be of special interest to diagnostic laboratories with a high sample turnover. A comparison of these three commercially available tests has not been published so far. In this study, the Kainos ELISA assay most reliably differentiated study groups (Table 2, Figure 2). Additionally, its results were the most representative of renal function parameters (Table 3). The calculated intra- and inter-assay CVs for precision assessment were all <15%, which is consistent with previous studies assessing the Kainos ELISA assay [34] and indicates good test performance. 

As the stability of FGF-23 in feline serum samples stored at −20 °C was demonstrated for a storage period of up to 28 days [36] and 14 days [29], the FGF-23 concentrations in our study were likely not affected by storage at −20 °C for a maximum of 5 working days. Although the laboratory recommends taking blood samples in cats that have fasted overnight, it was not possible to collect information regarding the diet and fasting status of the cats included due to the retrospective study design. Therefore, a potential influence of postprandial state and diet composition on the concentrations of the parameters of renal dysfunction and FGF-23 concentrations is possible. Correlations between FGF-23 concentrations, and the renal parameters and phosphorus in this study, therefore, need to be judged with caution and were mainly made to compare the performance of the three assays tested.

The results of this study also highlight the importance of establishing individual assay-based reference intervals for FGF-23 concentration. To the authors’ knowledge, 56–700 pg/mL is the established universal reference interval for FGF-23 concentration measurements in geriatric cats with a median age of 13 years using the Kainos ELISA assay [31]. This reference interval is much broader than the one established for healthy humans (8.2–54.3 pg/mL) [39]. Another study in cats proposed a reference range of 0–366 pg/mL for the Kainos ELISA assay [34].

An association between increasing FGF-23 levels and CKD progression has been documented in cats, as well as humans and dogs [29,40,41,42]. FGF-23 is considered a potential parameter for early CKD diagnosis and monitoring [31]. None of the three assays were able to clearly differentiate between study groups I and II, or II and III, which suggests that FGF-23 may be suboptimal for the early detection of CKD or the monitoring of early-stage kidney disease. However, clinical information about the cats, as well as results from urinalysis, were lacking, precluding IRIS staging. Conclusions regarding the clinical role of FGF-23 as a diagnostic marker, therefore, are beyond the scope of this study.

All three assays consistently differentiated study groups I and II from study groups III and IV. Especially the results from the Kainos ELISA assay were significantly different between study groups I and II, I and IV, and II and IV; therefore, the assay was able to distinguish between cats with mild and severe azotemia.

Creatinine is an indirect biomarker of the GFR. FGF-23 concentration is known to be inversely related to the GFR in humans [43] and cats [31]. Therefore, the strong and very strong correlations observed with the Kainos ELISA and DiaSorin assays between FGF-23 and creatinine concentrations could be expected. However, the results of the MyBioSource ELISA assay only moderately correlated with creatinine levels. The association between FGF-23 and SDMA is well documented in both dogs and cats [29,31,35,40,41,42]. Interestingly, the Kainos ELISA assay showed a moderate correlation with SDMA, and the correlation was low according to the DiaSorin assay. The MyBioSource ELISA assay showed no correlation between FGF-23 and SDMA levels (Table 3). Both of these findings call to question the value of measurements taken using this particular assay.

Hyperphosphatemia secondary to CKD has been associated with shorter survival times in both cats and humans [43,44,45]. It has been suggested that even early stages of CKD lead to dysregulation of phosphate homeostasis in cats [31,32]. In human medicine, increased FGF-23 concentrations precede hyperphosphatemia in patients with CKD [27]. FGF-23 might be, therefore, an attractive potential tool for therapeutic decision regarding phosphorus control in early kidney disease in cats. In our study, FGF-23 concentration measured using the Kainos ELISA assay showed a moderate correlation with phosphorus levels. The correlation was low for the results of the DiaSorin assay, and no correlation at all was found with the MyBioSource ELISA assay (Table 3). As FGF-23 is a phosphaturic hormone, some correlation with increasing phosphate concentrations is to be expected for all three diagnostic assays. A previous study in cats diagnosed with CKD that used the MyBioSource ELISA assay reported no correlation between the levels of FGF-23 and phosphorus in serum samples [35]. This is consistent with our study results. The results of this study indicate that the choice of assay can influence the measurement results and thus possibly diagnostic decision making. Further studies are necessary for the confirmation of this hypothesis.

Overall, the value of FGF-23 as a biomarker remains unclear. Findings in previous studies indicate its potential for monitoring CKD rather than establishing a diagnosis. This study highlights the importance of considering the method for FGF-23 analysis when interpreting test results and indicates that results retrieved with MyBioSource ELISA may not be relied upon. An automated assay such as the DiaSorin assay may be much more convenient in a commercial laboratory setting. Although, in this pilot study, it shows inferior performance compared with Kainos ELISA, it might have some potential in the future.

## 5. Limitations of the Study

One limitation of this study was its retrospective design. Potentially important background information was unavailable to the authors, including living conditions, diet, and the reason for blood sampling and laboratory testing. Therefore, no data regarding history, clinical signs, urinalysis, or diagnostic imaging were included in the study, and the exclusion of pre- or post-renal azotemia was not explicitly possible. However, the blood results of the cats chosen for this study did not show the other abnormalities usually seen in pre-renal (i.e., increased concentrations of total protein, albumin, sodium and/or chloride, and increased ratio of urea to creatinine) or post-renal (hyperkalemia) azotemia. Therefore, the authors considered azotemia to be renal in origin. In addition, the definition for staging feline CKD according to the IRIS warrants the confirmation of stable azotemia. As these were leftover samples, this information was lacking as was any information about history, clinical signs, imaging findings, blood pressure, and urinalysis. Specifically, the identification of an IRIS stage 1 group (renal disease present but not yet azotemic) is impossible without further diagnostic information. For these reasons, the cats included in this study were grouped into study groups I to IV rather than into IRIS stages 1 to 4. Most importantly, study group I was not synonymous with IRIS 1, and cats might not have had early kidney disease but likely represented healthy cats, especially as this group mainly consisted of young cats. However, considering the lack of a statistically significant difference between study groups I and II, this may confirm previous suspicions that FGF-23 is of limited value as an early biomarker of CKD. An additional limitation was the low number of animals in each study group.

## 6. Conclusions

The highest correlation with renal parameters and various levels of creatinine according to the IRIS was observed with the Kainos ELISA assay, which demonstrated intra- and inter-assay CVs of <15%, indicating good reproducibility for FGF-23 measurement in cats. All three diagnostic assays measuring FGF-23 concentrations can differentiate between mild and severe azotemia, but the relevance of FGF-23 as an early biomarker for detecting CKD remains unclear. FGF-23 may have some value for monitoring feline CKD. However, taking into consideration the correlation analysis with creatinine, SDMA, and phosphorus, this seems to be dependent on the diagnostic assay chosen. Individual assay-based reference values should be established to make a reliable interpretation of FGF-23 levels possible to diagnose or monitor feline CKD.

## Figures and Tables

**Figure 1 animals-13-01853-f001:**
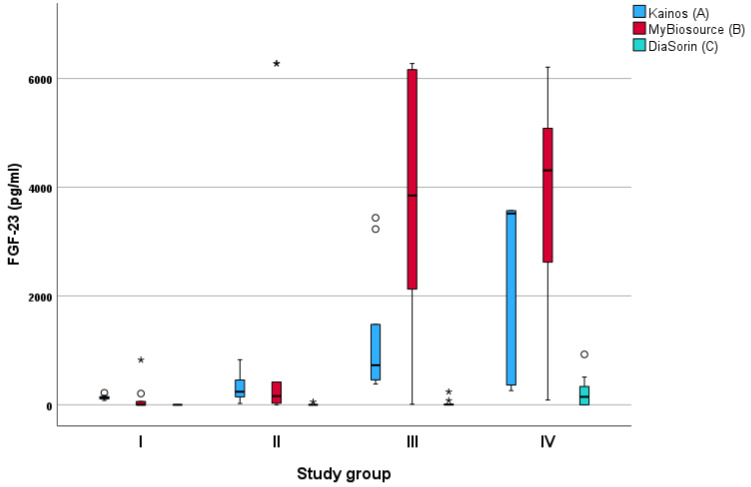
Measurements of Fibroblast growth factor (FGF)-23 concentration using three different diagnostic assays in cats divided in study groups I–IV by creatinine concentration according to the guidelines of the International Renal Interest Society (IRIS). ° = mild outliers (values that were more than 1.5 × interquartile range below Q1 or above Q3 in the boxplot); * = extreme outliers (values that were more than 3.0 × interquartile range below Q1 or above Q3 in the boxplot). (A) Kainos = FGF-23 ELISA Kit, Kainos Laboratories, Tokyo, Japan; (B) MyBioSource = MyBioSource, San Diego, CA, USA; (C) DiaSorin = LIAISON FGF 23 kit for intact FGF-23 on the Liaison platform developed by DiaSorin (Saluggia, Italy).

**Figure 2 animals-13-01853-f002:**
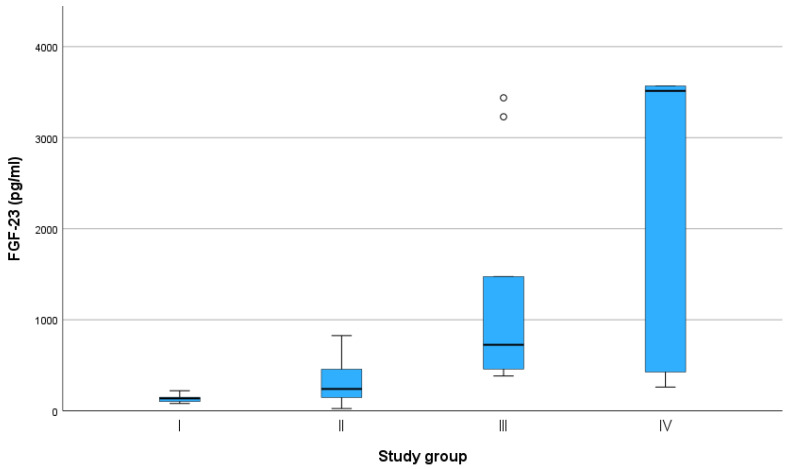
Fibroblast growth factor (FGF)-23 concentrations measured using the FGF-23 ELISA Kit (Kainos Laboratories, Tokyo, Japan) in cats classified in study groups I–IV by creatinine concentration level according to the guidelines of the International Renal Interest Society (IRIS). ° = mild outliers (values that were more than 1.5 × interquartile range below Q1 or above Q3 in the boxplot).

**Table 1 animals-13-01853-t001:** Age, biochemical parameters, and Fibroblast growth factor (FGF)-23 concentration in cats when divided in groups I–IV by creatinine value according to the guidelines of the International Renal Interest Society (IRIS) (N tested cats, mean (M), median (ME), standard deviation (SD), minimum (Min), and maximum (Max)).

Study Group	RI	I	II	III	IV	*p* ^1^
Age (years)	-	N = 6	N = 7	N = 8	N = 8	-
		M = 1.3	M = 14.6	M = 13.4	M = 11.3	
		ME = 1.0	ME = 15.0	ME = 15.0	ME = 10.5	
		SD = 0.5,	SD = 1.62	SD = 4.2	SD = 4.1	
		Min = 1.0,	Min = 12.0	Min = 6.0	Min = 5.0	
		Max = 2.0	Max = 17.0	Max = 18.0	Max = 16.0	
FGF-23 (pg/mL) ^A^	56–700 ^2^	N = 10	N = 10	N = 10	N = 10	χ^2^ = 26.095, *df* = 3, ***p* < 0.001**
		M = 133.3	M = 301.9	M = 1266.1	M = 2566.6	
		ME = 136.3	ME = 239.9	ME = 725.5	ME = 3513.0	
		SD = 42.4	SD = 251.1	SD = 1146.2	SD = 1544.9	
		Min = 80.6	Min = 24.5	Min = 382.9	Min = 260.7	
		Max = 220.9	Max = 826.4	Max = 3437.5	Max = 3567.8	
FGF-23 (pg/mL) ^B^	56–700 ^2^	N = 9	N = 10	N = 10	N = 10	Χ^2^ = 17.805, *df* = 3, ***p* < 0.001**
		M = 121.1	M = 1373.2	M = 3654.6	M = 3840.5	
		ME = 0.0	ME = 159.4	ME = 3845.9	ME = 4311.1	
		SD = 271.7	SD = 2587.0	SD = 2220.3	SD = 2164.6	
		Min = 0.0	Min = 0.0	Min = 11.7	Min = 88.8	
		Max = 822.4	Max = 6275.4	Max = 6275.4	Max = 6275.0	
FGF-23 (pg/mL) ^C^	56–700 ^2^	N = 10	N = 10	N = 10	N = 8	Χ^2^ = 13.164, *df* = 3, ***p* = 0.004**
		M = 0.0	M = 5.3	M = 35.0	M = 321.5	
		ME = 0.0	ME = 0.0	ME = 3.6	ME = 155.7	
		SD = 0.0	SD = 16.8	SD = 76.0	SD = 381.4	
		Min = 0.0	Min = 0.0	Min = 0.0	Min = 0.0	
		Max = 0.0	Max = 53.3	Max = 240.0	Max = 924.1	
Creatinine (µmol/L)	<168 ^3^	N = 10	N = 10	N = 10	N = 10	Χ^2^ = 36.596, *df* = 3, ***p* < 0.001**
		M = 74.9	M = 209.5	M = 325.9	M = 863.4	
		ME = 78.5	ME = 209.0	ME = 307.0	ME = 633.5	
		SD = 19.4	SD = 25.5	SD = 50.6	SD = 452.2	
		Min = 29.0	Min = 179.0	Min = 265.0	Min = 455.0	
		Max = 94.0	Max = 247.0	Max = 403.0	Max = 1549	
SDMA (µmol/L)	<0.75 ^3^	N = 2	N = 10	N = 9	N = 10	Χ^2^ = 21.410, *df* = 3, ***p* < 0.001**
		M = 0.55	M = 0.81	M = 1.33	M = 3.16	
		ME = 0.55	ME = 0.80	ME = 1.27	ME = 2.66	
		SD = 0.04	SD = 0.25	SD = 0.49	SD = 1.91	
		Min = 0.52	Min = 0.42	Min = 0.66	Min = 1.11	
		Max = 0.58	Max = 1.26	Max = 2.42	Max = 7.33	
Phosphorus (mmol/L)	0.8–1.9 ^3^	N = 10	N = 10	N = 10	N = 10	Χ^2^ = 21.671, *df* = 3, ***p* < 0.001**
		M = 2.0	M = 1.4	M = 1.9	M = 4.5	
		ME = 2.0	ME = 1.4	ME = 1.7	ME = 3.5	
		SD = 0.5	SD = 0.3	SD = 0.8	SD = 2.4	
		Min = 1.2	Min = 0.9	Min = 1.1	Min = 2.2	
		Max = 2.7	Max = 1.8	Max = 3.2	Max = 8.9	
Urea (mmol/L)	5.0–11.3 ^3^	N = 10	N = 10	N = 10	N = 10	Χ^2^ = 34.083, *df* = 3, ***p* < 0.001**
		M = 8.3	M = 17.9	M = 29.6	M = 70.3	
		ME = 7.8	ME = 15.5	ME = 28.0	ME = 66.8	
		SD = 2.7	SD = 5.6	SD = 8.8	SD = 27.8	
		Min = 5.4	Min = 12.1	Min = 17.8	Min = 43.0	
		Max = 13.4	Max = 25.9	Max = 47.4	Max = 131.8	
Calcium (mmol/L)	2.3–3.0 ^3^	N = 10	N = 10	N = 10	N = 10	Χ^2^ = 0.748, *df* = 3, *p* = 0.862
		M = 2.4	M = 2.5	M = 2.5	M = 2.5	
		ME = 2.5	ME = 2.5	ME = 2.5	ME = 2.5	
		SD = 0.2	SD = 0.1	SD = 0.2	SD = 0.2	
		Min = 2.0	Min = 2.3	Min = 2.2	Min = 2.2	
		Max = 2.8	Max = 2.6	Max = 3.0	Max = 2.8	
Sodium (mmol/L)	145–158 ^3^	N = 10	N = 10	N = 10	N = 10	Χ^2^ = 1.384, *df* = 3, *p* = 0.709
		M = 152.6	M = 153.3	M = 153.6	M = 154.9	
		ME = 151.5	ME = 154.0	ME = 153.0	ME = 153.0	
		SD = 4.6	SD = 2.7	SD = 2.2	SD = 5.0	
		Min = 144	Min = 150.0	Min = 152.0	Min = 151.0	
		Max = 161	Max = 157.0	Max = 158.0	Max = 166.0	
Potassium (mmol/L)	3.0–4.8 ^3^	N = 10	N = 10	N = 10	N = 10	Χ^2^ = 4.300, *df* = 3, *p* = 0.231
		M = 4.8	M = 4.3	M = 4.0	M = 4.7	
		ME = 4.9	ME = 4.3	ME = 3.8	ME = 4.5	
		SD = 0.7	SD = 0.6	SD = 0.8	SD = 1.2	
		Min = 3.5	Min = 3.1	Min = 2.8	Min = 2.8	
		Max = 5.8	Max = 5.1	Max = 5.3	Max = 6.9	

FGF-23 = Fibroblast growth factor-23; RI = reference interval; SDMA = symmetric dimethylarginine. ^1^ Statistical differences among study groups I–IV, Kruskal–Wallis test with Bonferroni correction, where *p* < 0.05 indicated statistical significance. ^2^ Reference range based on Finch et al. (2013) [31]. ^3^ Reference ranges based on the internal values of the LABOKLIN laboratory (Bad Kissingen, Germany). ^A^ FGF-23 ELISA Kit, Kainos Laboratories, Tokyo, Japan; ^B^ MyBioSource, San Diego, CA, USA; ^C^ LIAISON FGF 23 kit for intact FGF-23 on the Liaison platform developed by DiaSorin (Saluggia, Italy). Values in bold represent statistical significance (*p* < 0.05).

**Table 2 animals-13-01853-t002:** Comparison of Fibroblast growth factor (FGF)-23 concentration measurements performed using three different diagnostic assays among individual study groups based on creatinine concentration according to the staging of the International Renal Interest Society (IRIS).

Study Groups	Kainos ^A^	MyBioSource ^B^	DiaSorin ^C^
I–II	*p* = 0.975	*p* = 0.401	*p* = 1.000
I–III	***p* = 0.001**	***p* = 0.002**	*p* = 0.209
I–IV	***p* < 0.001**	***p* = 0.002**	***p* = 0.008**
II–III	*p* = 0.124	*p* = 0.452	*p* = 0.554
II–IV	***p* = 0.010**	*p* = 0.342	***p* = 0.031**
III–IV	*p* = 1.000	*p* = 1.000	*p* = 1.000

^A^ FGF-23 ELISA Kit, Kainos Laboratories, Tokyo, Japan; ^B^ MyBioSource, San Diego, CA, USA; ^C^ LIAISON FGF 23 kit for intact FGF-23 on the Liaison platform developed by DiaSorin (Saluggia, Italy). *p* < 0.05 indicated statistical significance. Values in bold represent statistical significance (*p* < 0.05).

**Table 3 animals-13-01853-t003:** Correlations among parameters involved in renal dysfunction and three different Fibroblast growth factor (FGF)-23 diagnostic assays using Spearman’s rank correlation coefficient.

		Crea	P	Ca	SDMA	Na	K	Urea	FGF-23 ^A^	FGF-23 ^B^	FGF-23 ^C^
Crea	CC	-	**0.521 ****	0.109	**0.907 ****	0.161	−0.046	**0.950 ****	**0.764 ****	**0.510 ****	**0.950 ****
	Sig	-	**<0.001**	0.504	**<0.001**	0.321	0.778	**<0.001**	**<0.001**	**0.001**	**<0.001**
	N	40	**40**	40	**31**	40	40	**40**	**40**	**38**	**38**
P	CC	**0.521 ****	-	0.268	**0.687 ****	0.189	**0.373 ***	**0.563 ****	**0.532 ****	0.209	**0.420 ****
	Sig	**<0.001**	-	0.095	**<0.001**	0.252	**0.018**	**<0.001**	**<0.001**	0.208	**0.009**
	N	**40**	40	40	**31**	40	**40**	**40**	**40**	38	**38**
Ca	CC	0.109	0.268	-	0.033	0.193	**0.341 ***	0.169	0.260	0.103	0.163
	Sig	0.504	0.095	-	0.859	0.232	**0.031**	0.298	0.105	0.537	0.327
	N	40	40	40	31	40	**40**	40	40	38	38
SDMA	CC	**0.907 ****	**0.687 ****	0.033	-	0.196	0.079	**0.829 ****	**0.580 ****	0.319	**0.429 ***
	Sig	**<0.001**	**<0.001**	0.859	-	0.291	0.674	**<0.001**	**<0.001**	0.086	**0.020**
	N	**31**	**31**	31	31	31	31	**31**	**31**	30	**29**
Na	CC	0.161	0.189	0193	0.196	-	-0.093	0.280	**0.332 ***	0.251	**0.337 ***
	Sig	0.321	0.242	0.232	0.291	-	0.568	0.081	**0.036**	0.128	**0.039**
	N	40	40	40	31	40	40	40	**40**	38	**38**
K	CC	-0.046	**0.373 ***	**0.341 ***	0.079	-0.093	-	0.007	-0.136	-0.196	-0.212
	Sig	0.778	**0.018**	**0.031**	0.674	0.568	-	0.966	0.403	0.238	0.202
	N	40	**40**	**40**	31	40	40	40	40	38	38
Urea	CC	**0.950 ****	**0.563 ****	0.169	**0.829 ****	0.380	0.007	-	**0.835 ****	**0.624 ****	**0.572 ****
	Sig	**<0.001**	**<0.001**	0.298	**<0.001**	0.081	0.966	-	**<0.001**	**<0.001**	**<0.001**
	N	**40**	**40**	40	**31**	40	40	40	**40**	**38**	**38**
FGF-23 ^A^	CC	**0.764 ****	**0.532 ****	0.260	**0.580 ****	**0.332 ***	-0.136	**0.835 ****	-	**0.560 ****	**0.730 ****
	Sig	**<0.001**	**<0.001**	0.105	**<0.001**	**0.036**	0.403	**<0.001**	-	**<0.001**	**<0.001**
	N	**40**	40	40	**31**	**40**	40	**40**	**40**	**38**	**38**
FGF-23 ^B^	CC	**0.622 ****	0.209	0.103	0.319	0.251	-0.196	**0.624 ****	**0.560 ****	-	**0.423 ***
	Sig	**<0.001**	0.208	0.537	0.086	0.128	0.238	**<0.001**	**<0.001**	-	**0.010**
	N	**40**	40	40	31	40	40	**40**	**40**	38	**38**
FGF-23 ^C^	CC	**0.510 ****	**0.420 ****	0.163	**0.429 ***	**0.337 ***	-0.212	**0.572 ****	**0.730 ****	**0.423 ***	-
	Sig	**<0.001**	**0.009**	0.327	**0.020**	**0.039**	0.202	**<0.001**	**<0.001**	**0.010**	-
	N	**40**	**40**	40	**31**	**40**	40	**40**	**40**	**38**	38

Ca = calcium; CC = correlation coefficient; Crea = creatinine; FGF-23 = Fibroblast growth factor-23; Na = sodium; K = potassium; P = phosphorus; SDMA = symmetric dimethylarginine; Sig = statistical significance. ^A^ FGF-23 ELISA Kit, Kainos Laboratories, Tokyo, Japan; ^B^ MyBioSource, San Diego, CA, USA; ^C^ LIAISON FGF 23 kit for intact FGF-23 on the Liaison platform developed by DiaSorin (Saluggia, Italy). * Correlation was significant at the 0.01 level (2-tailed); ** correlation was significant at the 0.05 level (2-tailed). Values in bold represent statistical significance (*p* < 0.05).

## Data Availability

Not applicable.

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
