# Peer review of "Comparison of Three Different Diagnostic Assays for Fibroblast Growth Factor-23 (FGF-23) Measurements in Cats: A Pilot Study"

_animals, 2023, doi:10.3390/ani13111853_

Round 1
Reviewer 1 Report
Lines 31-32: Correlations between assays measuring the same biological parameter are not very helpful – the authors should use Bland-Altman analysis to report level of agreement and bias rather than reporting correlation coefficients.
Line 55: As FGF23 is renally excreted, it is also influenced by GFR in the cat (see Finch et al., 2013)
Line 68-69: The precise mechanisms regulating FGF23 release into the circulation are poorly understood. Acute increases in phosphate intake leading to post-prandial hyperphosphataemia do not necessarily lead to immediate rises in serum FGF23 concentration although chronic increases in dietary phosphate intake are associated with increases serum FGF23 concentrations. Thus the relationship between serum phosphate and FGF23 is more complicated than indicated here.
Line 103 to 106 – IRIS stages are indicated by Arabic rather than Roman numerals.
Line 132-133: This sentence does not make sense ‘SDMA levels could be measured for 31 cats, the remaining in all 40 cats.’
Was information gathered on whether the samples from the 40 cats were collected following overnight fast? This information is important in interpreting the laboratory tests including phosphate which can change in the immediate post-prandial state depending on the type of diet fed.
Did the authors look at transforming the data (for FGF23 in particular), prior to statistical analysis? Ln transforming FGF23 often seems make the data normally distributed and easier to present and analyse by statistical methods that are able to account better for multiple comparisons between groups.
Figure 1 and associated methods – as the authors do not present any clinical data (history or physical exam findings) they should make it clear that grouping these cats based on plasma creatinine concentration should not be confused with staging these cats. In order to be staged, cats need to have had a diagnosis of CKD made by the clinician who should use history and physical exam findings together with laboratory test results (persistence of elevated creatinine and association with USG <1.035 would be often used) in coming to such a diagnosis. In addition, in order to stage cats by IRIS staging system they need to be assessed as being stable by the clinician examining them. None of these data appear to be available on the 40 cases used in this study. This needs adjusting throughout the manuscript.
The formatting of table 1 makes it difficult to read.
Lines 190-196: As stated earlier, looking at the agreement between the assays using Bland-Altman analysis would be more informative than the strength of correlation – a Bland Altman plot could also be used to determine where the differences between the assay lay (what concentration range) – i.e. where in the range of assay concentrations any biases are found.
Lines 221-222: As indicated above, the authors should not equate the groups of cats used in this study with IRIS stages as they did not have enough information from each cat to stage the cat’s kidney disease. There are other places in the manuscript where this is implied.
Lines 224-226: The concentrations that were used for low, moderate and high concentrations in the intra- and inter-assay CV evaluations should be stated in the manuscript.
Line 271-272: FGF23 is also inversely related to GFR in cats (Finch et al., 2013).
Line 286-288: If FGF23 increases before plasma phosphate increases in mineral bone disorder of CKD it is not surprising that FGF23 correlation with phosphate is not strong, particularly in early stage CKD where plasma phosphate has not increased yet. Phosphate overload fills up intracellular stores prior to plasma phosphate increasing and increase in FGF23 secretion is an adaptive response in early stage CKD to maintain phosphate excretion such that it balances phosphate intake despite a reduction in GFR. For these reasons it is not surprising that the correlation between FGF23 and phosphate is only moderate. Furthermore, if samples times were not standardised in terms of time of feeding, this will have led to some cats having post-prandial rises in plasma phosphate which would complicate the relationship. The authors need to accept this limitation of their study.
Conclusions: The authors should accept that their study design was not adequate to draw any conclusions about the value of FGF23 as a biomarker for diagnosis or monitoring of early stage CKD and so they should not draw any conclusions on this. Furthermore, as there was no gold standard for the measurement of FGF23 with which each of the assays evaluated could be compared, the authors rely on expected biological performance of each assay in cats with elevated serum creatinine. The lack of other patient data to enable robust classification of these cats in terms of the type of kidney disease they were suffering from is a real limitation of this study. Really the main conclusion they can draw is that the different assays are not giving equivalent results and the My Biosource assay does not perform in a way that would be expected from the biology of FGF23.
Reasonable quality
Reviewer 2 Report
Please find suggestions for the authors in the attached file
